# Specificity and Origin of the Stability of the Sr Isotopic Ratio in Champagne Wines

**DOI:** 10.3390/molecules26165104

**Published:** 2021-08-23

**Authors:** Robin Cellier, Sylvain Bérail, Julien Barre, Ekaterina Epova, Anne-Laure Ronzani, Cornelis Van Leeuwen, Stanislas Milcent, Patrick Ors, Olivier F. X. Donard

**Affiliations:** 1Institut des Sciences Analytiques et de Physicochimie Pour l’Environnement et les Matériaux, UMR CNRS 5254, Hélioparc, 2 Avenue P. Angot, 64053 Pau, France; sylvain.berail@univ-pau.fr (S.B.); anne-laure.ronzani@univ-pau.fr (A.-L.R.); 2MHCS, 20 Avenue de Champagne, 51200 Epernay, France; smilcent@moethennessy.com (S.M.); pors@moethennessy.com (P.O.); 3Advanced Isotopic Analysis, Hélioparc, 2 Avenue P. Angot, 64053 Pau, France; julien.barre@ai-analysis.com (J.B.); ekaterina.epova@ai-analysis.com (E.E.); 4EGFV, Bordeaux Sciences Agro, INRAE, ISVV, Université de Bordeaux, 33882 Villenave d’Ornon, France; vanleeuwen@agro-bordeaux.fr

**Keywords:** ^87^Sr/^86^Sr, Champagnes, musts, blending, Sr concentration

## Abstract

The ^87^Sr/^86^Sr ratio of 39 Champagnes from six different brands, originating from the whole “Appellation d’Origine Contrôlée” (AOC) Champagne was analyzed to establish a possible relation with the geographical origin. Musts (i.e., grape juice) and base wines were also analyzed to study the evolution of the Sr isotopic ratio during the elaboration process of sparkling wine. The results demonstrate that there is a very homogeneous Sr isotopic ratio (^87^Sr/^86^Sr = 0.70812, n = 37) and a narrow span of variability (2σ = 0.00007, n = 37). Moreover, the Sr concentrations in Champagnes have also low variability, which can be in part explained by the homogeneity of the bedrock in the AOC Champagne. Measurements of the ^87^Sr/^86^Sr ratio from musts and base wines show that blending during Champagne production plays a major role in the limited variability observed. Further, the ^87^Sr/^86^Sr of the musts were closely linked to the ^87^Sr/^86^Sr ratio of the vineyard soil. It appears that the ^87^Sr/^86^Sr of the product does not change during the elaboration process, but its variability decreases throughout the process due to blending. Both the homogeneity of the soil composition in the Champagne AOC and the blending process during the wine making process with several blending steps at different stages account for the unique and stable Sr isotopic signature of the Champagne wines.

## 1. Introduction

The geographical origin of food products has become an increasingly important issue for consumers, producers and distributors. The precise knowledge about the origin of a product in relation to a specific production area is now considered as a tool to create added value. The opportunity to trace the geographical origin is now most often, a priority for products that owe their reputation of being produced from a specific region or «terroir» such as premium wines. Linking wines to their «terroir» of origin is a certification approach increasingly applied to wines. Scientists are therefore challenged to establish unambiguous parameters to identify the geographical area of grape harvest and wine production.

The huge diversity of production areas represents a real challenge to secure the provenance of wine. The wine composition is influenced by many factors such as soil, climate, grapevine variety, training system, vineyard floor management and other parameters linked to oenological practices and the use of additives [1,2]. In this context, for all wines, and more specifically here for sparkling wines, the Sr isotopic ratios can be considered as an important variable that will trace the “terroir” and the environment of the wine. Among different inorganic compounds, the level of Sr contained in wines directly originates from the vineyard soil. A series of studies demonstrate the continuous and conservative signature of 87Sr/86Sr ratios in the system soil–grapes–wine [3,4,5,6,7,8]. The specificity of the Sr isotopic ratio has also been used to trace wines and identify the vineyards which they originated from around the world. This ratio(87Sr/86Sr) has been applied for the identification of the geographical origin of wines from South Africa [8], Argentina [6], Canada [7], Portugal [3] and Romania [9]. The 87Sr/86Sr value is influenced by the age and composition of bedrocks underlying the vineyards [10]. In general, strontium is actively taken up by both plants and animals due to its similarities with calcium [11]. However, during the uptake of Sr and the metabolism pathway of the plant, the Sr isotopic ratio is unaffected so its signature is extremely conservative [5]. It remains generally unchanged throughout the process from grapes to wine [12,13].

In this study, we have followed the evolution of Sr isotopic ratio during the whole winemaking process of Champagne, from the fruit to the final sparkling wine and to fully address the final Sr isotopic signatures in a wide array of Champagne brands. The different steps of the Champagne process were investigated to monitor the potential evolution of the Sr isotopic signatures. Grapes, musts, base wines and final products were analyzed. First, the Sr isotopic ratios were analyzed in the musts, the base wines and the Champagnes originating from six major and distinct brands produced with grapes originating from the whole AOC Champagne. Second, the Sr isotopic ratios were followed in greater detail on samples originating from two specific small plots (around one hectare each, mono varietal grapevine) to better understand the evolution of Sr isotopes during the elaboration process at a small scale. These specific brands from single vineyards do not result from any blending. This study allows the establishment of a link between the final product and its territory of origin for future authentication of Champagne and later identification of counterfeited products.

## 2. Materials and Methods

### 2.1. Champagne Winemaking Process and Samples Collection

General Sr isotopic signatures of traditional Champagne products: The different critical steps of Champagne elaboration are presented in Appendix A. Grapes are harvested at the beginning of September. The grapes are then gently pressed to expel the juice and to obtain the must (grape juice). The must is then pre-blended and left in the tank during the alcoholic fermentation for two to three weeks to obtain the base wines. The base wines are then blended to give away to the desired “cuvées”. Then the “cuvée” is bottled with the addition of the yeast and the sugar necessary for the second alcoholic fermentation. This second fermentation takes place in the bottle and lasts for several months. After this fermentation, the bottles are stored with the lees composed of dead yeast and bentonite for one to up to seven years. Then, the yeast, bentonite, and other compounds that have settled at the neck of the bottle are frozen in liquid nitrogen and expelled naturally during the disgorgement step by the pressure. The bottle is then topped up and corked to be ready for shipping. To obtain a representative sample set of isotopic Champagne signatures, a collaboration with six Champagne houses has been established. With their support, a large array of samples has been collected at all critical steps of the wine making procedure. The important and most critical step in the Champagne making process is the blending procedure that is performed to generate a stable and distinctive taste for each brand. The pre-blending consists of mixing musts coming from different sub-areas of the AOC Champagne (between 10 and 20 musts) to obtain the base wines. The types of musts and the proportion of each in the pre-blended base wine are implemented under the supervision of the oenologist (winemaker) in charge of the production. The blending is expected to have an impact on the evolution and stability of the Sr isotopic ratio in Champagnes.

The first part of the study was applied to the entire Champagne AOC. Samples were collected from the six different brands of Champagne. The recovered base wines and Champagne samples were respectively pre-blended and blended. The number of distinct Champagne samples is summarized in the Appendix A. The samples are:-Must samples: During the 2018 harvest, 25 must samples were recovered from areas all over the AOC Champagne. Must is the juice originating from the grapes after a gentle pressing of the grape. Stalks, skins and pips were removed immediately. Each of the 25 musts were collected from a specific and delimited wine growing area from the AOC Champagne and each must be from only one grapevine variety (Chardonnay or Pinot noir or Meunier).-Base wines: In 2019, 11 base wines produced only from the Marne department were collected. The base wines are obtained after the pre-blending of several musts.-Champagne samples: 39 Champagnes from six different brands from vintages ranging between 1983 up to 2016, white and rosé, vintages and non-vintages, brut and dry were collected and analyzed. These 39 Champagnes were all elaborated with grapes produced in the AOC Champagne. The Champagne is the final product obtained after blending the base wines.

Sr isotopic signatures of Champagne originating from a single and specific geographical area: After unraveling the evolution of Sr isotopes on Champagnes with the primary products originating from different areas of the AOC Champagne, we have examined specific Champagnes brands originating only from the same small limited geographical location collecting samples directly from the grape to the final product. These specific brands are not blended and are sold as “vintage” every year, they originate from two small parcels producing “single vineyard” Champagne. These two parcels are also planted with a single grapevine variety; one is composed of 100% Chardonnay and the other of 100% Pinot noir. Therefore, the results from these samples will be the same, starting from the grape to the final Champagne product, without any addition or blending of grapes or musts from different geographical origins (Appendix A). The only impact on the evolution of the Sr isotopic ratio, if any, will result only from the origin and the process of elaboration. These specific samples recovered from the two plots are:-Parcel A: 1 kg of grapes from Chardonnay, one sample of must from the same variety, a base wine from the same must and a Champagne from the same base wine.-Parcel B: 1 kg of grapes from Pinot noir, one sample of must from the same variety, a base wine from the same must and a Champagne from the same base wine.

### 2.2. Reagents and Standards

All sample preparations prior to analysis were performed in an ISO 7 cleanroom. All dilution steps involved in the preparation of standards, samples, and reagents for elemental and isotopic analysis were performed with 2% HNO_3_ (67–70%, ULTREX^®^ II Ultrapure Reagent, J.T.Baker, Fisher Scientific, Illkirch, France) and were obtained by dilution with Ultrapure de-ionized water (18 MΩcm^−1^) from a Milli-Q analytical reagent grade water purification system. For the wine sample digestion H_2_O_2_ (30%, ULTREX^®^ II Ultrapure Reagent, Fisher Scientific, France) and HNO_3_ were used in Savillex vials. The pre-concentration of samples was performed by evaporation in 15 or 30 mL PFA Savillex vials (Savillex Corporation, Eden Prairie, MN, USA) under a laminar fume hood. The sample evaporation was performed with a closed evaporation device (Evapoclean, Analab, France).

An “in house” reference material of Champagne (ChRM) was also analyzed in triplicate during each session to assure the quality of the sample mineralization and of the analysis [14]. The specific analytical strategy and the Champagne reference material were elaborated to yield highly stable and reproducible analytical conditions over several years on different three ICP-MS multicollector systems [14]. A strontium isotopic standard SRM 987 (pure SrCO3, NIST, Gaitherburg, MD, USA) was used as quality control for each session of Sr isotopic analysis. The NIST SRM 987, certified for its Sr isotopic composition with an 87Sr/86Sr certified value of 0.710255 ± 0.00023 was used for the bracketing procedure. A general series of samples analysis and bracketing strategies were developed to obtain excellent intercomparing results between the different analytical sessions over 3 years [14].

### 2.3. Sample Preparation

For the Sr isotopic analyses, the samples were first preconcentrated by evaporation and the samples were then digested. After complete digestion, the samples were then purified on a Sr selective chelation column (Triskem). For the Sr matrix purification and the selective rubidium removal, the columns of ion-separation (2 mL, Triskem International, Bruz, France) were filled with 120 mg of Sr selective resin Eichrom^®^ (SR-B50-S). The resin washing step was performed twice with the addition of 5 mL of Ultrapure 3 M HNO_3_ and then followed by the addition of 5 mL of Ultrapure water five consecutive times. The conditioning of the columns was performed with 2 mL Ultrapure 3 M HNO_3_ then the sample was added to the column (4 mL in Ultrapure 3 M HNO_3_). The elution matrix was realized with 4 mL of Ultrapure 3 M HNO_3_ and rinsed again twice. The final elution of the Sr isotopic analytes was obtained after rinsing these columns twice with 5 mL of H_2_O. After purification, the samples were analyzed with the different MC-ICP-MSs after adjusting the solutions to yield optimal precision [14].

### 2.4. Instrumentation and Operating Conditions

The analysis of Sr isotopes was performed on several MC-ICP-MS (Nu Plasma, Nu Plasma HR and Nu 1700) under wet plasma conditions. The parameters for Sr isotopic analysis were optimized using the standard solution NIST SRM 987 with a concentration of 150 µg·L^−1^ or 200 µg·L^−1^ depending on the relative sensitivity of the MC-ICP-MS used. The typical signal for the isotope ^88^Sr was about 7 V (Appendix A). All analyses were performed using a sample introduction mode using a self-aspirating micro concentric nebulizer (200 µL·min^−1^) combined with a glass cyclonic spray chamber. A conventional sample–standard–bracketing calibration sequence with the NIST SRM 897 was used as standard bracketing. The instrumental blank was subtracted using the On Peak Zero approach (OPZ), the ^87^Sr/^86^Sr ratio was corrected for mass bias using the constant ratio ^86^Sr/^88^Sr of 0.1194, and from the potential remaining interferences from traces of ^87^Rb using the ratio ^85^Rb/^87^Rb of 2.5926. A second correction was then applied to the bracketing standard in agreement with the study of Albarede et al., 2004. The value of ^87^Sr/^86^Sr ratio for NIST SRM 987 applied for data processing was 0.710255, according 15.

## 3. Results and Discussion

### 3.1. Sr Isotopic Signatures in Champagne Brands and over Time

#### 3.1.1. Sr Isotopic Ratio in Six Different Champagne Brands

The Sr isotopic ratios of 39 Champagnes from six different brands were determined by MC-ICP-MS. The harvest year of these samples is either the year of the vintages (in the case of vintage Champagne) or the predominant year of harvest of the “cuvée”. The time span of the different Champagne analyzed ranged from 1983 up to 2016 (Figure 1).

The results from Figure 1 demonstrate that there is remarkable homogeneity in the Sr isotopic ratios among the different Champagne brands from the AOC area, as well as over time. The Sr isotopic average of all the samples is ^87^Sr/^86^Sr = 0.70812 (±0.00007, 2 s, n = 39). The samples only display a slight variation of the Sr isotopic ratio between 0.708075 and 0.70823. The Sr isotopic ratio can be related to the overall homogeneity of the geology, homogenous nature of the soil in the AOC Champagne and the Champagne making process.

Several studies have addressed this question and did not reveal any differences in the Sr isotopic signatures during the different stages of wine production. [3,12,13,15,16,17,18,19,20,21,22,23]. However, in very few specific cases, mostly dealing with white wine productions, the use of filtration and adjuvants or the removal of specific parts of the grape (skin, pips and stalks) have been reported to slightly alter the final Sr isotopic composition of the product [16]. In these studies, only still wines are considered, and the case of other sparkling wines was not investigated. For sparkling wines, the elaboration process is quite different compared to still wine. First, the skins and seeds are removed at the very beginning of the process. The juice of the grapes is then recovered by a gentle pressing of the grapes. This procedure removes a significant amount of the organic and inorganic compounds located in the solid parts of the grapes (pips and skin). These will be lost and will not appear in the final wine. These steps could also potentially alter the Sr content and its isotope composition.

Finally, it is also most remarkable to mention that ^87^Sr/^86^Sr are also extremely homogeneous through time (over almost 40 years). This observation relevant to time is consistent with the findings of Braschi, E: From Vine to Wine: Data on 87Sr/86Sr from Rocks and Soils as a Geologic and Pedologic Characterization of Vineyards, 2018, [17] who stated that no variations of the Sr isotopic signature could be observed for Italian wines harvested over four different years. In the case of still wines from prestigious chateaux of the Bordeaux area, only very minor variations of Sr isotopic ratios could be observed over a 10 year span [18]. It, therefore, appears that the climatic conditions of the vintage only marginally affect the uptake conditions of the Sr from the soil to the plant.

This unique Sr isotopic signature of Champagne wines (Figure 1) is compared against the range of Sr isotopic ratios of other sparkling and still wines from around the world (Figure 2). It is worth mentioning the extremely narrow span of the Sr isotopic composition, related to both the homogeneity of the soil and the blending process. It is indeed much narrower than all major European sparkling wines (European wines are between 0.7070 and 0.7120; Champagne between 0.70805 and 0.70818). It is interesting to notice that the wines from Québec display a much higher Sr isotopic ratio as well as a much larger span of signatures. These observations can be certainly related to the fact that the Quebec soils are overlying on an old granitic bedrock. It is then most likely that in this very old bedrock, the Rb has decayed to Sr over time [7]. When examining the span of the Sr isotopic ratio of the Prosecco, we can observe that is also very large (between 0.7075 and 0.7120) and this has also to do with the underlying bedrock. Indeed, the bedrock of the different wine regions of Italy are very different and massively impacted by the formation of the mountain chains The Alps, bringing up a wide array of different bedrocks. These diversified bedrocks certainly translate in the topsoil and should explain the variation of Sr isotopic ratio of wine from Lambrusco, Aglianico and Cenanese. The span of Bordeaux wines is also large (0.7080 to 0.7105) [24]. This situation is also due to the varieties of the bedrocks in the region which originate from the Oligocene era to the Pliocene era [18]. The stability of the Sr isotopic ratio of Champagne linked to the homogeneity of the bedrock of the Champagne region is clearly observed.

#### 3.1.2. Unravelling the Stability Factor of Sr Isotopic Ratio Signatures in Champagne

##### Geogenic Sr Isotopic Signature in the Musts

The musts are directly obtained from the gentle direct pressing of the grapes. They are the first stage of the different important steps of the Champagne elaboration. As musts have not been affected by the different processing steps that occur later in elaboration, their Sr isotopic ratio will reflect most closely with the geogenic signature from the soil. The entire AOC Champagne is located within the sedimentary Parisian Basin. The AOC area represents 33,000 hectares reflecting the similar nature of soils located on a large homogenous outcrop of the Parisian sedimentary basin. The vineyards of the AOC Champagne are located on homogenous bedrocks in terms of geology. The soil composition is also rather homogenous in Sr isotopic ratio in the production area. Champagne vineyards are located on chalk and dolomite originating from the Upper Cretaceous or Cenozoic areas in the Marne department and the Upper Jurassic in the Aube department. The Sr isotopic ratios measured in these geological formations ranges between 0.7075 and 0.7079 [21] which is in agreement with those obtained in all of the whole Champagne samples (0.70812). Figure 3 clearly displays that all the must samples have been collected either in the Marne department at the border of the Cenozoic and Upper-Cretaceous outcrops. All the other southern musts samples are located mainly in the Upper Jurassic outcrops. The median and the range of the Sr isotopic signatures of the musts in close relation with their geographical localization of the must’s origin is presented as boxplots in Figure 3b. The distribution of the musts over the isotope ratio range is presented in Figure 3b,c. The Sr isotopic ratios of the musts are slightly less homogenous than those obtained for the final Champagne products. Nevertheless, the musts are relatively narrow (min= 0.70898 and max = 0.70832) and the average of Sr isotopic ratio of the must is ^87^Sr/^86^Sr = 0.70817 (±0.00022σ, n = 25) while for the Champagne it is ^87^Sr/^86^Sr= 0.70812 (0.00007, 2σ, n = 39). The averages are quite similar, but the standard deviations are smaller for Champagne due to the blending.

The first important factor controlling this remarkable Sr isotopic ratio in the final Champagne produced certainly lies with the similarity of the nature of the topsoil. Most of the grapes used for the white Champagne are produced in the department of the Marne (≈70%). The Aube produces approximately 20% and the Aisne 10%. The AOC Champagne is located on bedrocks mainly composed of chalk and dolomite and extends over lower and Upper Cretaceous and Cenozoic outcrops [22]. The musts are the closest in isotopic signature to the one from the soil while the Champagne integrates the entire vinification procedures. Figure 3b presents the very small dispersion of the Sr isotopic ratios for the musts originating from the different areas, presenting only minimal average discrimination between the musts originating from the soils on top of the Cenozoic formations with those of the Upper Cretaceous, both of them being mainly evenly distributed between the Aube and Marne department. Statistical analysis was made with Matlab. The analysis of variance (ANOVA one way) was performed on the results presented in Figure 3b. The results obtained show that there is no significant difference of Sr isotopic value between the musts produced on soils from Jurassic, Cretaceous or Cenozoic bedrock with a significance (*p* < 0.05). Figure 3c shows that the blending process of the different musts already yields a median Sr isotopic signature that is very close to the final commercial bottle signature of the Champagnes.

Impact of the Grape-Wines on the Sr Isotopic Signatures in the Musts: In the AOC of Champagnes, several grapevines varieties are planted and are used to produce the musts and base wines for the different brands of Champagne. The varieties most commonly used, in the AOC champagne, are Chardonnay, Pinot noir and Meunier. Sr isotopic ratios of musts as a function of grapevine variety are presented in Figure 4.

The results presented in Figure 4 clearly display the fact that the grapevine variety does not impact the Sr isotopic signatures in the musts. All grapevine varieties behave similarly with respect to the uptake and translocation of the Sr and its isotopic signature from the soil to the plant and finally the grapes. Each grapevine variety absorbs the Sr in a similar way with apparently no significant fractionation. Similar results are reported by Di Paola-Naranjo et al., 2011 for Cabernet Sauvignon, Malbec and Syrah grapevine varieties.

The Blending is a major cause of the homogeneity of Sr isotopic ratios in musts and Champagne. The effect of blending on the evolution of the Sr isotopic ratio during the wine making process was closely followed. In order to characterize this potential effect in detail, 13 base wines from the same year of harvest (2018) and from the same department (Marne), where most of the Champagne vineyard are located, were analyzed. There are two blending steps during the Champagne making process, which is one of the particularities of Champagne. A first blending of the musts (pre-blending) is performed to obtain the base wine, and a second blending of the base wines takes place later to obtain the final Champagne.

The effect of blending on the Sr isotopic ratio from must to Champagnes is presented in Figure 5. The effect of the different blending steps can be clearly seen in Figure 5 with the statistical analysis and the standard deviation that decreases through the elaboration process (from 0.00022 for must to 0.00007 for Champagne).

The Sr isotopic signature of must, which is closest to the original signature from the soil, ranges from 0.7088 to 0.8034. This composition range is slightly higher than that of both the base wines and the Champagne, in which the ratio ranges from 0.70805 to 0.70818. These results clearly demonstrate the importance and impact of the different blending operations, resulting in a very homogenous Sr isotopic signature in the final product. We performed and analysis of variance (ANOVA one way) to test for the significance of the results obtained and displayed in Figure 5. The ANOVA results show that there are no significant differences in Sr isotopic ratios between the musts and final Champagne products and also between the base wines and final Champagne products in case of significance test with *p* < 0.05. On the contrary, a significant difference can be observed between Sr isotopic value between must and base wine. This observation is logical and can be explained by the sampling of the base wine, which is only from the Marne department.

The blending process used in Champagne mixes musts from different bedrocks. Hence, the biochemical Sr signature can be compared for musts from various origins and Champagne by plotting the Sr isotopic ratio as a function of the concentration of Sr (1/[Sr]) as presented in Figure 6. Despite the large variation of Sr in the soils (10 to 1000 mg/kg) [23], the overall isotopic Sr signatures in the musts do only vary in a very small range between 0.70780 and 0.70830.

These results are obtained when produced on the similar calcareous formations of the Marne, Aube and Aisne departments. The Champagne Sr isotopic signatures are close to the signature of the musts produce in the Marne department and present an even much more limited span in both the Sr concentrations (175 to 350 µg/kg) as well as in the Sr isotopic ratios (0.708059 to 0.708183). Indeed, 70% of the grapes used in most Champagnes brands are sourced from the Marne department.

These results highlight the homogeneity and the conservative behavior of the Sr isotopic ratio from the soil to the primary base wines as well as in the final products in the Champagne. The Sr isotopic ratios can present very slight variations when determined during the wine making process from the grapes and the soil but the overall signature is globally retained. These slight differences are then smoothed out later during the wine making process involving the numerous steps in the vinification chain and the large time span between the different procedures. The great homogeneity of the Sr isotopic ratio of the products can be explained at first by the low variability of this ratio in the geological bedrocks in the AOC Champagne. These findings are in complete agreement with the results from Braschi et al., 2018 [25], who found that that the Sr isotopic ratio of must and wines is related to that of the labile part of the soil. The available Sr is absorbed from the labile part of the soil by the vine roots without modification or fractionation of the Sr isotopic ratio. For the assessment of the provenance of the Italian wine “Cesanese” from six selected vineyards in the Lazio region, the ratio ^87^Sr/^86^Sr was determined in wines, musts, soils and rocks [13]. These authors report that the isotopic ratio of Sr remains. Another excellent correlation of the Sr isotopic ratio between geological substrates, vine shoots and grape juice was also evidenced earlier by [7,17,25].

It has also been demonstrated that during the absorption, there is no fractionation of the Sr by the plant [10,24]. This conservative mirror image in the Sr isotopic ratio from the soil to the plant has been previously observed. Indeed, it is now well-known that Sr isotope ratios of wines are linked to the labile fraction of the soil where the vines have grown [3,8,13,16,17,19,25,26]. Further to the homogeneity of the soils, the overall elaboration process does not, then significantly affect the average value of Sr isotopic ratio from the musts to the final products but results in a much narrow signature both in Sr concentration and in Sr isotopic ratios. In Braschi et al., 2018 [25], the Sr isotopic ratio of the labile fraction of the soil was similar to those recorded in the plant and the resulting wine.

### 3.2. Evolution of Sr Isotopic Ratio in Champagnes Produced from Specific Single Vineyards

#### 3.2.1. Sr Concentration from Grape to Champagne in the Specific Vineyards

This part of the study was performed on the two specific parcels (separated only by several kilometers) where each plot is planted with a single grapevine variety and where no blending is implemented during the Champagne elaboration process. One plot is growing the variety of Chardonnay and the other is planted with Pinot noir. These are two of the three varieties used in the AOC Champagne. Therefore, in this specific study, the modification of the ^87^Sr/^86^Sr will be limited only to the effects of the grape variety eventually and the soil. The specificity of this study is that the Sr isotopic ratios were determined and followed from the grapes up to the last stage of the Champagne making process.

Prior to the determination of the Sr isotopic ratios, the concentration of Sr for each sample was measured. The results are presented in Figure 7. The Sr concentration is much higher in the grapes than in the must, the base wine, and the Champagne. The most remarkable decrease takes place when the must is recovered from the grapes. This significant decrease can be explained by the fact that skins and seeds are removed during the pressing and that only the grape juice (must) is recovered. Hence, a large part of grape Sr content, which is located in the skin and pips, is removed from the must. Once the must has been obtained, the Sr concentration remains unchanged, after the elimination of the skin and pips, regardless of the elaboration stages and the grapevine variety. Without blending, the Sr concentration remains stable between must, base wine and Champagne. Similar concentrations of Sr are observed between the two terroirs for the musts, base wines and Champagne. The Sr concentrations observed are very close to other wines when compared with other wines (Milićević et al., 2018; later by Epova et al., 2019). The Sr concentration of Champagne wines is close to those of Bordeaux wines (around 0.3 mg·L^−1^) and is lower compared to the Sauvignon blanc and Cabernet Sauvignon (~2.3 mg·L^−1^).

#### 3.2.2. Evolution of the Sr Isotopic Ratio from Grape to Champagne in the Specific Vineyards

The results of the Sr isotopic ratios of grapes, must, base wines and Champagne, in Figure 8, show that the total inorganic Sr concentrations are different between the two vineyards, a difference can also be observed in the Sr isotopic ratio. The Sr isotopic ratios in the must, the base wine and the Champagne are on average 0.70815 in parcel A and 0.70785 in parcel B and according to the measurement conditions these values are significantly different although being quite close geographically. Since there is no blending at all during the different steps of the wine making process for these single vineyard Champagnes, and that we have previously demonstrated in this paper that the grape varieties have no influence on the Sr isotopic signature, these slight but significant differences of Sr isotopic signature most likely originate from the soil composition. These values are in agreement with the results reported by Willmes et al., 2018 since the ^87^Sr/^86^Sr isotopic ratios from chalk and dolomite in French soils (i.e., those found in the AOC Champagne) range between 0.7079 and 0.7095. These results highlight the conservation of the Sr isotopic ratio of Champagne during the wine making process and its direct reflection of the soil isotopic signature.

A slight trend of decrease in Sr isotopic composition can also be observed in both blocs between the grapes and the musts presented but then Sr signatures remain constant throughout the rest of the process (Figure 8). These slight differences should be linked to the removal of skins and pips during pressing, which could slightly influence the Sr isotopic ratio. Even after a loss in concentration of Sr between grapes and musts of a factor of 40 for parcel A (Chardonnay) and a factor of 20 for parcel B (Pinot noir), the Sr isotope ratios remain quite unchanged. The elaboration process does not significantly change the Sr isotopic composition of grapes/musts/base wines/Champagnes. These results are consistent with previously published data on other types of wines [12,13,16,17,19,20,27,28,29]. The Sr isotopic ratio of the Champagne is in general very similar to the Sr isotopic ratio of the labile fraction of the soil where the vine was grown [3,8,13,16,17,19,25,26].

The small differences in Sr isotopic signature that can be observed between these two parcels could reflect the difference in Sr isotopic signature of the soil as well as a possible impact of the grapevine variety. We have previously reported early in this paper that the grape varieties do not influence the isotopic signature. These findings have also been confirmed by several authors that have clearly demonstrated that the grapevine variety does not influence the Sr isotopic ratio [6,7]. Therefore, the difference of the Sr isotopic ratio of these two plots should be related to the difference of the Sr isotopic signatures of the soils even if their location is quite close to one another [13,17,19,21,25].

These results are very important since they also show that similarly to what has been observed on Bordeaux wines with Sr isotopic signature [30], when no blending takes place, we can discriminate the specific origin related to a small portion of land. The geogenic signature Champagne can be discriminated within the AOC Champagne areas if the blending and general Champagne making procedure is not applied for the elaboration of the final product. These observations are paramount when using this approach to fight against fraud and counterfeiting

## 4. Conclusions

For authentication and traceability of sparkling wines, Sr isotopic analyses of Champagnes were carried out. In this study, 39 samples of Champagnes from six different brands were analyzed by MC-ICP-MS. The results showed that the Sr isotopic composition of Champagne was very homogeneous in space (samples originating from the whole AOC area) and in time (samples sourced from vintages spanning a period from 1983 to 2016). No difference can be made between the six Champagne brands. The isotopic ratio of Sr is extremely stable for the 39 samples of Champagnes (2σ = 0.00007) making traceability easy. It appears that this stability is due mostly to homogeneous bedrocks in Champagne soil.

The Sr isotopic ratios of some musts and base wines were also analyzed. This investigation showed that the great stability of the ^87^Sr/^86^Sr ratio in Champagne was also due to the blending of musts and base wines during the elaboration. These blendings allow the averaging out of the slight differences between Sr isotopic ratio in musts and therefore decrease the variability observed in Champagne samples. It also appeared that the musts could not be differentiated according to their variety or by their department of origin, probably because the bedrock of the Champagne area is too homogeneous.

Analysis throughout the Champagne elaboration shows that the value of the Sr isotopic ratio is not influenced by the winemaking process. The Sr isotopic ratio of the final product is similar to the signature of the must and therefore linked to the soils and bedrocks on which the grapes were produced. Moreover, the blending allows the homogenization of the musts values, resulting in its high stability. This very small range of Sr isotopic ratio of Champagne is particularly precious information for traceability purposes.

## Figures and Tables

**Figure 1 molecules-26-05104-f001:**
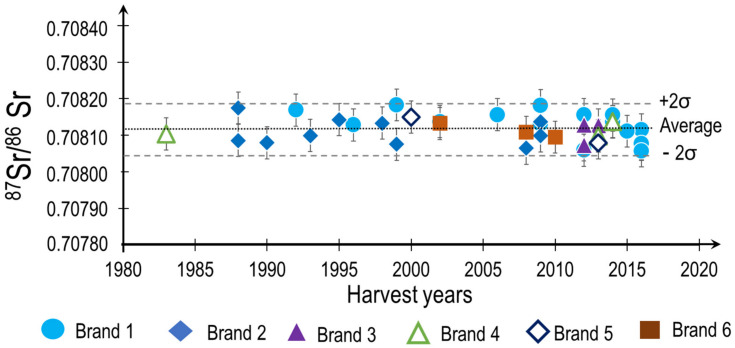
^87^Sr/^86^Sr as a function of the year of harvest of the 39 Champagnes from six different brands with the average and 2σ. The error bars (2σ) were obtained from a triplicate of the same sample.

**Figure 2 molecules-26-05104-f002:**
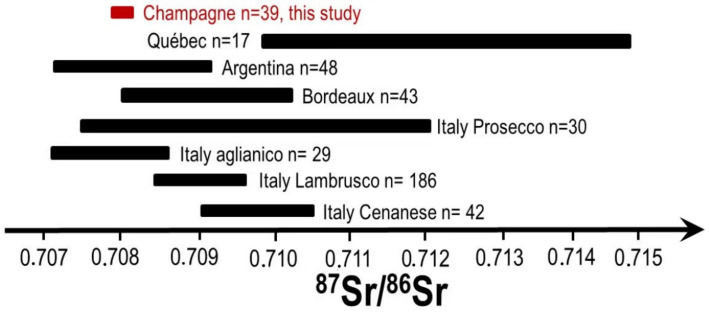
Values of Sr isotope ratio of some sparkling wines and still wines from around the world. Di Paola-Naranjo et al., 2011, Durante et al., 2018, 2013, Epova et al., 2019, Marchionni et al., 2013, 2016, Petrini et al., 2015 and Vinciguerra et al., 2016.

**Figure 3 molecules-26-05104-f003:**
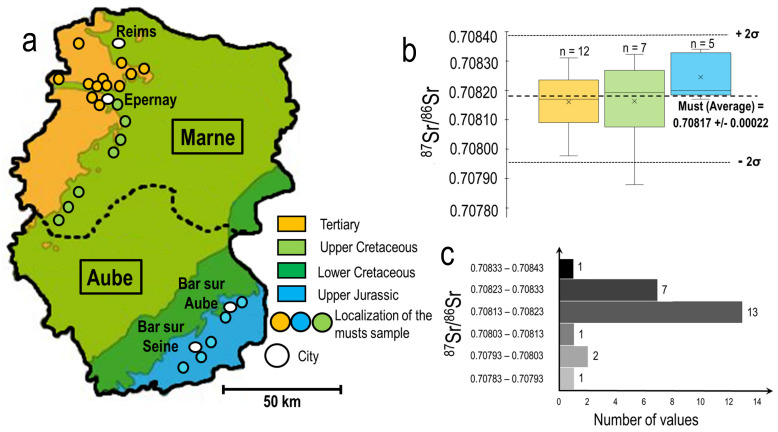
(**a**) Projection of the musts samples as a function of the type of bedrock, (**b**) associated Sr isotopic ratios measured in musts, (**c**) the distribution of the ^87^Sr/^86^Sr values of the musts.

**Figure 4 molecules-26-05104-f004:**
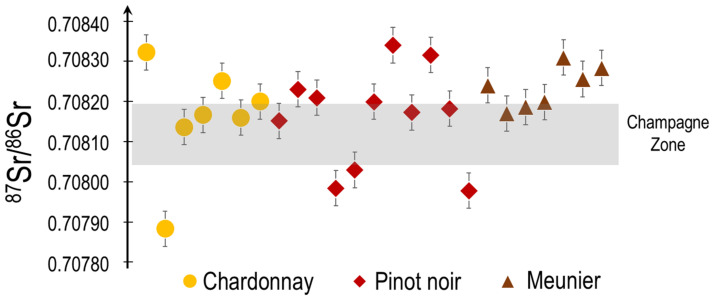
^87^Sr/^86^Sr ratio of musts produced from three grapevine varieties in Champagne the average ratio in Champagne ±2σ. Errors bars (2σ) are from triplicate analyses.

**Figure 5 molecules-26-05104-f005:**
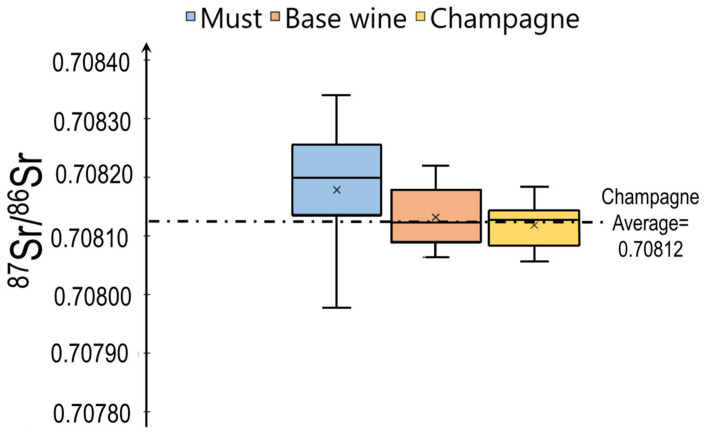
Evolution of the Sr isotopic ratio and the dispersion of the values (2σ) during the Champagne making processes.

**Figure 6 molecules-26-05104-f006:**
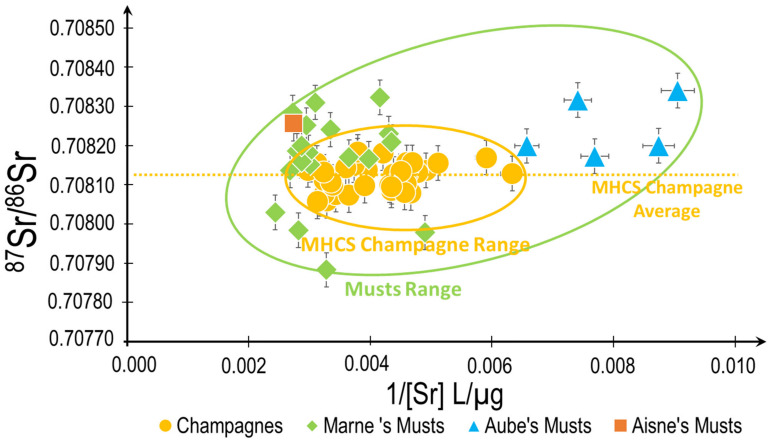
Sr isotopic biogeochemical mixing model applied to musts produced at specific locations in the Champagne AOC area and Champagnes.

**Figure 7 molecules-26-05104-f007:**
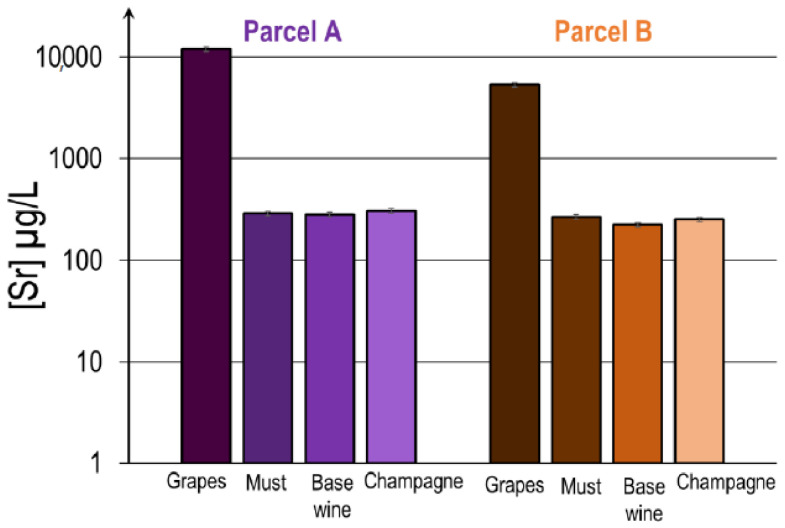
Evolution of the Sr concentration from grape to Champagne for two mono varietal parcels (A,B) (expressed in Log scale).

**Figure 8 molecules-26-05104-f008:**
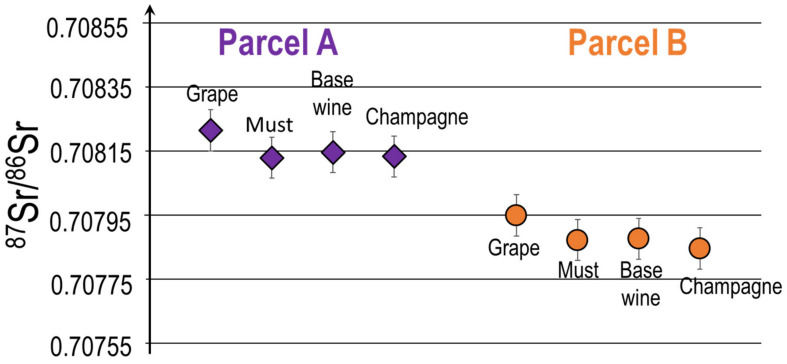
Evolution of Sr isotopic ratio from Grape to Champagne for two mono varietal parcels (A,B). Errors bars (2σ) are calculated on a triplicate of the same sample.

## Data Availability

The data used for the publication of this paper are available on request. They will be organized in an open data basis with a control after the agreement of the MHCS group.

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
