# Peer review of "Specificity and Origin of the Stability of the Sr Isotopic Ratio in Champagne Wines"

_molecules, 2021, doi:10.3390/molecules26165104_

Round 1
Reviewer 1 Report
First and foremost, it appears the figures (Fig. 1-8) are missing from the MS, which make reviewing it rather difficult. It looks like the supplemental figures were mistakenly included in the main MS. In this case, the review will go on, as best as possible, with the written content.
The authors set out to determine the stability and variability of the 87Sr/86Sr ratio within champagnes and the several steps in the vinification process using MC-ICP-MS. They measured the isotopic ratio and concentration within two different sets of champagne samples. The first set was of musts (pre-blended), base wines (blended), and then the final champagne product from 6 different brands. These samples represented several areas of the AOC Champagne region. The second set of samples was of a narrower and specific region whose samples were not blended, and therefore represent a Sr ratio that would be unimpacted by any sort of blending process (which presumably would average out any differences within the different regions where the grapes/musts/base wines are taken from). Ultimately such information could be used in validation processes to identify origins of champagnes and to identify counterfeit products.
The authors found that the Sr ratio is relatively stable across different brands and across time when measuring champagnes of different years, which suggests the Sr ratio is not largely influenced by changes in climatic conditions but rather more by the soil Sr content. They also found that the Sr ratio of the musts is relatively more variable than the final champagne product, attributed to the blending process. No differences were found between grape varieties. And overall the blending processes in the vinification is likely the cause for the more stable Sr ratio. For the samples taken from specific, single vineyards, the authors found that the biggest decrease in Sr concentration came between the steps of grapes to must, due to the removal of grape skins and pips containing much of the Sr content. But there was little change in the ratio in the elaboration of these samples to the final champagne product.
Overall the study appears sound and well carried out, but without the figures this is difficult to judge. These need to be added for a final MS. There are several grammatical errors through out the MS that need to be addressed, as well as potential formatting marks/errors (e.g. pg. 8, in the Sr ratios given). The findings that the Sr ratio resemble the AOC Champagne soils demonstrates the potential for using Sr ratios as a validation on wines, catching would-be counterfeits, adding to the significance of the MS.
Author Response
Dear Reviewer,
I apologize for the absence of figures that were on another file
Attached, the manuscript with the figures and the consideration of a majority of your comments
Thank you in advance
Sincerely

Reviewer 2 Report
The manuscript entitled "Specificity and origin of the stability of the Sr isotopic ratio in Chamgane wines" by Cellier et al. was revised. The subject is interesting for the scientific community. I consider the manuscript needs major revision. Authors will find some comments on the PDF file to work on before considered the manuscript for acceptance. Manuscript preparation was sloppy, and a bit frustrating since no opportunity to see the data either in figures or table was provided because Figures (except Fig 1) and data tables (Sr contents and isotope ratios) were missing. I wish I could see the data to revise it more consciously.

Author Response

(The authors gave the same response as above.)

Reviewer 3 Report
It is possible that the authors have presented a draft version of the manuscript, since there are no Figures 2-8 in it, so it is currently difficult to evaluate the manuscript as a whole.
However, there are the following comments to the submitted text:
- The Supplementary material completely duplicates Figure 1 and the tables from the manuscript.
- Often there are no punctuation marks in the text.
- Links to publications and the list of references are not designed according to the MDPI rules.
- Instead of Braschi et al. (reference number 4), it is better to refer to Braschi et al. "Tracking Sr-87/Sr-86 from rocks and soils to vines and wine: an experimental study on the geological and pedological characteristics of vineyards using a radiogenic isotope of heavy elements" in Science of the Total Environment, 2018.
The final conclusion on the publication of the manuscript will be possible to submit only after receiving the full version with figures.
Author Response

(The authors gave the same response as above.)

Round 2
Reviewer 1 Report
With the figures in place, the paper is in a much better position for review. Overall, as stated in the first round, the science appears sound and well carried out. The figures and data presented now make this clear. It’s clear that the Sr isotopic ratio is due mainly to the soil composition (or likely due to that), that the elaboration process does not change the ratio markedly, and that the variability in the Sr ratio decreases throughout the elaboration process due likely to the blending steps that occur with the musts and base wines. Also, interestingly, are the measurements from the specific parcel samples that have no blending steps and do not see this decrease in variability, but show a decrease in overall Sr concentration. However, there are a number of grammatical issues throughout the manuscript that should be addressed before publication. Below I’ve tried to detail several, but I would also recommend a thorough re-reading of the manuscript to fix and edit errors, and re-write certain sentences for better clarity and readability.
Abstract, sentence 2: Too long of a sentence, break it up for better clarity. Remove “overall very” (grammatically it’s a little odd, but it is also imprecise wording). “Sr isotopic”, I think the word ratio is missing after this? It’s also unclear in the sentence if by a “narrow span of variability” pertains to the isotopic ratio or the concentrations. It’s not clear what you mean by Sr concentrations in this sentence.
Abstract, sentence 3: Would reword this sentence, “Measurements of the 87Sr/86Sr ratio from musts and base wines show that blending during Champagne production plays a major role in the limited variability observed.”
Introduction, 2nd paragraph: In the sentence that reads, “However, the Sr isotopic ratio does not fractionate...” I take it to mean that by “fractionate” you mean to indicate that biological systems and metabolic pathways are not able to discriminate between one isotope from another, and so the ratio is unaffected passing through metabolism. I would recommend removing the word “fractionate” and rewording the sentence.
Section 2.1: Where it reads, “...Figure 1 on supplementary material.” I would rephrase this (and other instances throughout the paper referring to the supplementary material) to simply read, “Supplemental Figure 1.”
Figure 1: It looks like the y-axis labels might be text boxes? (Perhaps to replace , with . in the decimal place?) The authors may want to tidy up the figure before publishing (i.e. remove boxes around number, top number still has , and one label is misaligned) But otherwise this is a nice figure to show the samples used in the study related to their harvest year.
First paragraph following Figure 1, 1st sentence: Capitalize “Figure 1” (And do so in the other instances of the manuscript). Also this sentence should be reworded for better readability: “The results from Figure 1 demonstrate that there is remarkable homogeneity in the Sr isotopic ratios among the different Champagne brands from the AOC area, as well as over time.”
Third paragraph following Figure 1, 1st sentence: Missing the word “it” where it reads, “Finally, is also...” (insert before “is”). In the 3rd sentence where it reads, “...is consistent with the findings of 24 who stated...” Add the names of the study before the reference number.
First paragraph after Figure 2, lines 4-5: Where it reads, “It is indeed much narrower all major...” Add the word “than” before “all”.
Section 3.1.2.1, lines 25-27: Reword the changed sentence: “As musts have not been affected by the different processing steps that occur later in elaboration, their Sr ratio will reflect most closely with the geogenic signature from the soil.”
Lines 147-148, after Figure 6: I think the reference title was mistakenly applied, rather than the reference number?
Section 3.2.1, line 177: Where it reads, “These are two of the three varieties that are 3 and used...” Remove, “that are 3 and”.
First paragraph after Figure 7, lines 198-199: Here the authors state that the large decrease in Sr concentration seen between grapes and musts could lead to a change in the isotopic ratio, but that seems unlikely. From a biological standpoint, metabolism won’t (can’t) discriminate between different isotopes, so even when the bulk of skin and pips may have a larger concentration of Sr, that ratio of 87Sr/86Sr will be the same throughout the biological sample.
Section 3.2.2, line 213: I think instead of the word “been” the authors meant “being”.
Figure 8: While there is a clear difference in Sr isotopic ratio between the Parcel A and Parcel B samples, which the authors attribute to the soil Sr composition, what do the authors suggest is the reason for the decrease in the Sr isotopic ratio going from grapes to must, base wine, and champagne? If it were a matter of overall decreased Sr concentration, due to the removal of skin and pips, how could that affect the different isotopes in different ways?
Author Response
Dear Reviewer
Attached the revised manuscript.
I have taken into account all your comments. Thank You for your help with the English correction.

Reviewer 2 Report
The revised version of manuscript Molecules-1272748 still has some comments that were not taken into account, and I consider the authors important to have a better version of the article.
Attached you will find a pdf file with some comments in the second version of the manuscript.

Author Response
Dear Reviewer
Attached you can find the revised manuscript.
I have taken into account all your comments.
Thank you

Reviewer 3 Report
The authors have made significant changes to the manuscript, which allow it to be published in the journal Molecules.Author Response
Dear reviewer
Thank you for your comments
Attached the revised manuscript
